# Pharmacotherapy of infertility in Ghana: Why do infertile patients discontinue their fertility treatment?

**Stephen Mensah Arhin**[1]*, **Kwesi Boadu Mensah**[1], **Evans Kofi Agbeno**[2], **Diallo Abdoul Azize**[2], **Isaac Tabiri Henneh**[3], **Eric Agyemang**[4], **Charles Ansah**[1]

**1** Department of Pharmacology, Kwame Nkrumah University of Science and Technology (KNUST), Kumasi, Ghana, **2** Department of Obstetrics and Gynecology, School of Medical Sciences, University of Cape Coast, Cape Coast, Ghana, **3** Department of Pharmacotherapeutics and Pharmacy Practice, School of Pharmacy and Pharmaceutical Sciences, University of Cape Coast, Cape Coast, Ghana, **4** Department of Sociology and Social Work, College of Humanities and Social Sciences, Kwame Nkrumah University of Science and Technology, Kumasi, Ghana

* stephen.arhin@ucc.edu.gh

## Abstract

### Background

Globally, millions of people of reproductive age experience infertility. With that notwithstanding, most infertile patients undergoing pharmacotherapy withdraw from treatment before achieving the desired outcome. The reasons for their withdrawal, particularly in sub-Saharan Africa, have not been well examined, hence the need for this study.

### Objectives

The aim of the study was to examine why infertile patients discontinue pharmacotherapy prior to achieving conception.

### Methods

The study employed an exploratory qualitative design. Purposive sampling technique was used to recruit subjects into the study. Twenty infertile patients (fourteen females and six males) who discontinued their treatment, and eight attending health professionals who provided direct care to these patients were interviewed. Telephone and face-to-face interviews were conducted using a semi-structured interview guide. The data collected were transcribed, coded, and generated into themes using thematic content analysis.

### Results

The major reasons for discontinuation of infertility treatment included lack of support from male partners, seeking alternative treatment, unmet outcome, poor medical services, distance, stigmatization, and relocation.

**Data Availability Statement:** All relevant data are within the paper and its Supporting Information files.

**Funding:** The authors received no specific funding for this work.

**Competing interests:** The authors have declared that no competing interests exist.

## Conclusions

Patients and healthcare personnel shared both similar and diverse views on reasons for discontinuation of infertility treatment that reflect situations in a typical African setting, most of which are not reported in existing studies. The outcome of this study will provide insight for fertility therapists and policy makers in designing appropriate measures to facilitate maximum compliance and improvement in treatment outcome.

## Introduction

Within developed and developing nations, more than half of infertile couples seek medical care, and follow up on fertility assessment and evaluation [1]. Nevertheless, a significant percentage of couples decide to discontinue fertility care [2]. The prevalence of discontinuation among couples is estimated to be 34% [2], although discontinuation rates may differ across various societies [3].

Seeking treatment for infertility is a stressful situation that may be hampered by factors such as discomfort, complications and failure of medical methods for women [4], cost of treatment [5, 6], psychological burden [7], marital and personal problems [8, 9], perceived poor prognosis, age and parity [7, 10], education and home-to-hospital distance [11]. These prevailing problems mostly lead to appearance of impulsive behaviors, scattered wrath, depression, feelings of helplessness, worthlessness, incompetence, anxiety and negative beliefs towards themselves, which may further worsen the treatment seeking behaviors and outcome of treatment [7, 12, 13].

Additionally, Donkor et al. reported high incidence of lack of concentration, loneliness and decreased sexual satisfaction among infertile women [14]. On the other hand, men suffering from infertility have been found to experience loss of social status, verbal abuse and stigmatization [15].

Infertile patients undergo various forms of treatment, ranging from pharmacotherapy to assisted reproductive technologies such as *in vitro* fertilization (IVF). Most patients seeking infertility treatment in sub-Saharan African rely on drug treatment, probably due to low socio/economic status, and limited access to assisted reproductive technologies. Many existing studies in Western countries regarding discontinuation rates mainly concentrate on patients undergoing IVF [2, 7]. However, there is a paucity of studies on reasons for discontinuation by patients undergoing pharmacotherapy for infertility, particularly in sub-Saharan Africa. This is partly due to the fact that most fertility experts/researchers in the sub-region are concerned with positive treatment outcomes, and tend to ignore clients who may decide to stop treatment. Our previous study which assessed the outcome of pharmacotherapy among infertile couples revealed that approximately 19.4% of them conceived after going through drug treatment [16]. However, the retrospective nature of that study made it impossible to assess how clients complied with treatment and the reasons for non-compliance, particularly in the patients who had unsuccessful treatment outcomes. This calls for further study using a prospective approach. As such, between April 2019 and March 2021, a hospital-based survey from four different health facilities was conducted to assess the conception rate among infertile patients undergoing pharmacotherapy for infertility in the Cape Coast Metropolis, Ghana. The 482 patients recruited into the study were followed up for a minimum of 12 months. During the period of follow up, it was observed that some of them discontinued their treatment prior to achieving conception. In order to identify the obstacles such infertile patients

encounter during treatment, and to enable stakeholders to develop appropriate interventional strategies that will enhance quality of care, as well as meeting the changing needs of the patients, such reasons ought to be examined. It is against this backdrop that the current study was conducted to assess the reasons for discontinuation of fertility care among infertile patients undergoing pharmacotherapy for infertility in the Cape Coast Metropolis.

## Materials and methods

### Study design

The study employed an exploratory qualitative design to solicit views from patients and health-care professionals on the reasons for discontinuation of infertility treatment in Ghana.

### Setting

The study was conducted at four different fertility centers in the Cape Coast Metropolis in the Central Region of Ghana. These facilities were selected because they attend to most of the fertility cases within the sub-region and beyond. These facilities also receive referrals relating to fertility issues from health facilities within the Cape Coast Metropolis and from surrounding communities as well.

### Sampling

A total of 28 persons were interviewed in this study. These comprised twenty (20) infertile patients who discontinued their treatment (fourteen women and six men) and eight attending healthcare personnel who attended to the patients. The respondents were recruited using purposive sampling technique. Out of the eight healthcare professionals interviewed, four were nurses, and the remaining four were doctors. The health professionals who were actively involved in the management of the patients were purposively selected from each of the four health facilities. The socio-demographic factors of patients and healthcare professionals are represented in Tables 1 and 2 respectively.

**Table 1. Socio-demographic data of infertile patients.**

| Variables | Total | Gender (range or number) |
|---|---|---|
| Age Range | N/A | Males (29–52 years.); Females (24–43 years) |
| Discontinuation time | N/A | Males (1–3 months); Females (2–4 months) |
| **Occupation** | | |
| Trading | 6 | Male (1); Females (5) |
| Teaching | 4 | Males (2); Females (2) |
| Farmers | 4 | Male (1); Females (3) |
| Fitting Mechanic | 1 | Male (1); Females (0) |
| Professionals | 5 | Male (1); Females (4) |
| **Type of infertility** | | |
| Primary | 14 | Males (2); Females (12) |
| Secondary | 6 | Males (4); Females (2) |
| **Previous treatment** | | |
| Yes | 12 | Males (2); Females (10) |
| No | 8 | Males (4); Females (4) |

NA = not available.

**Table 2. Socio-demographic data of key informants.**

| Designation | Total | Gender (number) | Average yrs. of infertility care |
|---|---|---|---|
| Doctors | 4 | Males (3); Female (1) | Doctors = 14 years |
| Nurses | 4 | Males (0); Females (4), | Nurses = 10 years |
| **Total** | **8** | **Males (3); Females (5)** | N/A |

## Inclusion and exclusion criteria

Patients who complied with treatment and follow up reviews were excluded from the study. Included in the study were patients who skipped review appointments or discontinued treatment prior to achieving conception. Also included were healthcare professionals such as nurses and doctors who offered direct services to these patients.

## Data collection

Telephone interviews and Key Informant Interviews (KII) were used to collect the data from respondents. Data was collected by two of the principal investigators from the four fertility centers between September 1st to 31st December, 2021. With the permission of the respondents, interviews were recorded on phone for later transcription.

## Conducting of interviews

Telephone interviews were conducted for individual patients at their own convenient time and place using a semi-structured interview guide. The patients were contacted through telephone numbers that were collected on the first day of their visit to the clinic. The purpose of the call was explained to them, and if permission was granted, interviews were carried out. Eight key informant interviews were conducted. This included four nurses and four doctors who were directly involved in the care of the patients. Face-to-face interviews were conducted at the consulting rooms or office of the key informants using a semi-structured interview guide. The interviews were recorded electronically with the consent of the respondents and later transcribed verbatim.

## Assigning codes

To ensure the confidentiality of the respondents, the interview transcripts were assigned some serial codes. Thus, interviewees were labeled as Client interviewed Number one (CIN-1), Client Interviewed Number two (CIN-2), up to Client Interviewed Number twenty (CIN-20). Thus, the serial codes begin from CIN-1, CIN-2, CIN-3, CIN-4, and CIN-5. . . up to CIN-20). Concerning clinical staff, the first interviewee was labeled as Health Practitioner interviewed number one (HP-1), Health practitioner interviewed number two (HP-2), up to Health Practitioner interviewed number eight (HP-8). The outcome of the interview was generated into themes based on the research objectives and questions asked.

## Data analysis

The audio-recorded interviews were transcribed verbatim and analyzed using thematic content analysis [17]. To familiarize ourselves with the transcript, we ensured that it was read several times by the principal investigators for clarity. Two of the authors independently coded four (4) of the transcripts (two from the patients and 2 from the healthcare), and the research team compared the outcome to ensure uniformity and reliability of the findings. One of the principal investigators (the first author) then coded all the transcripts using the coding system

agreed upon by the research team. A codebook was generated based on the major teams of the study. After identifying the emerging themes and sub-themes, it was then written out in the results. The identified themes and subthemes were then supported with specific quotes from the respondents.

### Ethical approval

Ethical clearance was obtained from the Institutional Review Board (IRB) of the University of Cape Coast (Ref No: UCCIRB/EXT/2019), and the Ethical Review Board of the Cape Coast Teaching Hospital (Ref No: CCTHERC/EC/2019). An informed written consent was obtained from all the patients after explaining the objective of the study to them during the initial recruitment stage of the research.

## Results

Nine themes were identified from the data. These include husband unsupportiveness, unmet outcome, alternative treatment, financial challenges, poor services, distance, relocation, psychological distress, and stigmatization. Verbatim quotes from the patients and the key informants were used to support the themes and subthemes. The summary of the themes and subthemes obtained from the patients and from the key informant interviews are presented in Tables 3 and 4 respectively.

### Why do infertile patients discontinue treatment?

This section presents the qualitative findings on the issue of why some patients discontinue treatment or skip review appointments with clinicians.

**Table 3. Themes and subthemes from infertile patients.**

| No | Theme | Subthemes |
|---|---|---|
| 1 | Husbands' unsupportiveness | Unwillingness to do semen analysis |
| | | Men unconcerned because of extramarital affairs |
| | | Having children already |
| 2 | Unmet outcome | Problem still persisting |
| | | Delayed conception |
| 3 | Alternative treatment | Local herbal drugs |
| | | Homeopathic clinics |
| 4 | Financial challenges | Cost of lab investigations |
| | | Cost of drugs |
| | | Consultation fees |
| 5 | Poor services | Lack of satisfaction with care |
| | | Unavailability of experienced doctors |
| 6 | Distance | Lack of easy access to fertility centers |
| | | Partners not staying at the same place |
| 7 | Relocation | Work factor |
| | | Lockdown |
| 8 | Psychological distress | Loss of confidence |
| | | Constant state of worrying |
| | | Emotionally traumatized |

**Table 4. Themes and subthemes from key informant interviews.**

| No. | Themes | Subthemes |
|---|---|---|
| 1 | Husbands' unsupportiveness | Unwillingness to do semen analysis |
| | | Less concerned about the problem |
| 2 | Unmet outcome | Delayed outcome |
| | | Lack of expected results |
| 3 | Financial challenges | Cost of lab investigations |
| | | Cost of drugs |
| 4 | Distance | Fertility centers situated in cities and work on special days |
| | | Most travel from far rural areas |
| 5 | Stigmatization | Social isolation |
| | | Societal pressure |

## Husband unsupportiveness

With regard to why patients discontinue treatment or review appointments with clinicians, some of the respondents said their husbands do not adhere to treatment. This is typical of a society where women are mostly and unjustifiably held responsible for childlessness. Accounts of why most infertile patients discontinued treatment due to apathy on the part of the male partners were common. The women were mostly left stranded to seek their own care because the male partners see themselves as 'strong' and 'fit' to reproduce. For instance, CIN-2 disclosed the refusal of her husband to come for investigation, as she said:

*Is my husband oo, he did not want to come and do the test.*

CIN 11 also shared a similar view on her discontinuation of treatment by saying:

*When the doctor initially told my husband to come and do semen analysis, he said that, his sperms are okay.*

Again, most of the interviewees recounted that, their husbands do not want to adhere to treatment and come for diagnostic investigations because they already have children. A quote from CIN- 10 recounts:

*We are not divorced but is being a while that we saw each other. Is like he doesn't stay at one place. I have been trying that, things will work for me. As for him, he has children. That is it. I am the only person trying hard to get the solution.*

Similarly, CIN-11 reiterates:

*You know the men, when they are able to impregnate one or two women, they think that, everything about them is correct. He thinks that is my own problem and I am the cause so I should seek my own help.*

CIN-3 also shared a similar view when she disclosed:

*My husband said he will come but he is doing some style. He has also relaxed because he already has two children.*"

In a similar view, CIN-19 also recounted:

*My husband is always finding excuse to come with me. When I came the first day, the doctor wrote some labs for me to do. He informed me that, my husband has to come for some test. When I went home and I informed him about it, he said he will come but always finding excuses. I can see that; he does not care so much because he has a child already. It seems I am much concerned about the situation than my husband.*

Another woman (CIN-20) also reiterated how uncooperative her husband was although she does not seem to know why her husband was doing that.
She said:

*Hmmm. My husband is doing himself some way. As if he is not interested in the marriage anymore because of the problem. He shows less concern about seeking care. Now it seems I am giving up. Yes. Is a big challenge but what can I do*?

## Unmet outcome

Another dimension disclosed by the patients that influenced them to discontinue treatment was the lack of expected outcome. The key expectation of patients seeking infertility treatment is conception and live birth. Anything short of that is not tolerated as most of them do not want to waste their time.
CIN-1 narrates her ordeal as she said:

*I was not seeing any results. I came several times but the problem was persisting. When you start with expectation and you do not see the outcome, it becomes worrying.*

In a similar account, CIN-11 added:

*I have been coming severally but nothing is coming so I have closed my mind small. When the doctor gives me the drugs, my menses sometimes delay, thinking that I may be pregnant. But when I check for pregnancy, there is nothing there.*

CIN-12 and CIN-19 also shared similar views as they recounted a delay in achieving conception as the major reason for discontinuation of treatment.

## Alternative treatment

Another key reason that contributed to discontinuation of treatment was seeking of alternative treatment (e.g. local herbal medications). Responses from CIN-5, CIN-6, and CIN-19 suggested that they discontinued their visit to the clinics because they wanted to try complementary and alternative treatment. For instance, CIN-5 said:

*We went to herbal clinic. The medicine that they prescribed we wanted to finish with it before we consider going back to the hospital".*

CIN-6 also revealed:

*I stopped and went to Accra to see specialist (name of facility withheld). I went there too, as well as (name of facility withheld). I later came back for the specialist to see me again. I wanted to try their system too.*

The use of local herbal medications was another reason that was shared when CIN-1 said:

*I resorted to herbal medicine. When you start with expectation and you do not see the outcome, it becomes worrying. For that reason, I switched to herbal treatments.*

### Financial challenges

A major factor that also influenced some of the patients to discontinue treatment was financial difficulties. Most of the patients (CIN-4, CIN-10, CIN-12, CIN-13, CIN-14, and CIN-18) recounted their challenges with regards to finances. For example, CIN-4 disclosed:

*It is because of money that is why we stopped coming. We paid a lot of money the last time we came. We did a lot of investigations but doctor said he cannot identify what is wrong with me so we need to do other investigations somewhere. But before the investigation is done, we needed to pay GhȻ 1000 but we did not have the money. That is why we stopped coming although the problem is not solved.*

CIN-4 further noted with regards to the facility charges by saying:

*We have to pay for consultation fees even if we are going for review. They treat us like new patients all the time.*

Additionally, CIN-8 also complained about expensive drugs within the facility when he revealed:

*I had a little problem at the dispensary. The drugs. You see. The price at the facility is too much. It is something that put most of the customers off.*

In a similar manner, CIN-13 added:

*Hmmm. My problem has been there for quite some time now. I have spent a lot of money on it. But unfortunately for me, I am just a farmer and do not have too much money. I only come when I have been able to gather some enough money from my farm proceeds. When the money gets finished, I have to suspend my coming until I have been able to raise some money again. As we are speaking now, like you asked me, my time for review has passed but I don't have the means to come. I am even in the farm right now even though the problem is not solved.*

CIN-18 additionally shared his view on financial challenges when he said:

*My major challenge has been finances. We need to pay for services anytime we come. Besides, we come from a very far place, so coming with my partner requires a lot of money. Sometimes, when the medications are written for us, we are unable to buy all. That is our main reason for not coming.*

### Poor services

Other interviewees (CIN-5, CIN-8, and CIN-17) also recounted lack of satisfaction with care as the common reason for termination of treatment. For instance, CIN-8 recounted:

*They keep changing the doctors that come there. Today this person will come and they change him. Tomorrow this person will come and they change him too. You call the doctor and he respond 'I am in Accra'. You call this person and he says 'ooh I have been transferred'. We were not happy coming there again because errm one person sees your problem and works on it. But the doc who is taking care of us, we call him today I am not there, we call him and he says I am not there.*

CIN-8 additionally shared his view on lack of experienced doctors as the reason for discontinuation by saying;

*Is like try try try. Small children try try. The students and erh so you see the service is not all that good. You see, is like UCC students who attend to us so you will not get gynecologist who is responsible for such issues to attend to you the way you want. The students will be asking you some things like, you see, the things they ask is like noo. Is not all that convincing.*

Similarly, CIN-17 added his views on how change of doctors affected his compliance with treatment when he narrated:

*Initially I had conviction that, since is a big hospital, I will have better services. But sometimes when you come, you see that the services are not all that good. It did not meet my expectation. Sometimes you need to come as early as possible because of distance but after waiting for a number of hours, then you are told that the doctor will not come. So, you see, you become so disturbed. When you are lucky to have a new person, he may not be experienced as you wish. Yes. So that was my problem. It happened on two occasions that is why I decided not to come again.*

CIN-5 also shared another view when she questioned the treatment option that was given to her husband without providing additional alternatives. She said:

*We have not been coming because the option that they were giving my partner, is like I was not satisfied with it. That they may operate on him. So, I wanted to see if there is any other hospital that I will get another option like given of medications instead of the surgery to improve his situation.*

## Distance

Another factor that affected patients' cooperation with treatment was distance. Some of the patients stay far from the treatment centers, as they travel from the outskirts of the city to access the fertility clinic. CIN-18 recounted his challenges when he said:

*We come all the way from Praso so is not easy to follow all the orders they give us. That is not our wish though, but if for any reason, we delay a bit, then we have to suspend our coming because we may not meet the doctor when we get there late.*

In a similar account, CIN-14 said:

*My place is very far and I need to come early too. So sometimes when I get to know that I am late, I don't come at all. This is because when I come and they have finished receiving cards for that day, I have to go back without seeing the doctor.*

Another issue with regards to distance was when one of the partners stayed at a different place. This made it very difficult for them to come together when it was required. For instance, CIN-15 reiterated:

> *I came alone the first day. The doctor told me to bring my partner but he is not here. He works in different region and comes here some weekends. You know the hospital does not attend to us on weekends. I have decided to go there so that we can see a doctor at where my husband works.*

## Relocation

In other views, CIN-7 and CIN-9 attributed their discontinuation to lockdown or relocation to different region. CIN-7 indicated:

> *My review time happened to be in the lockdown period. More so I have relocated from Cape Coast to Kumasi. So apart from the lockdown, relocation also affected our ability to come for review again.*

CIN-9 also recounted similar by saying:

> *At the moment we are not even at Cape Coast anymore. But we will try and come again. we have moved away from Coast because of work.*

## Psychological distress

Most of the respondents were subjected to psychological distress which affected their treatment seeking behaviors. Some of the patients with psychological distress (CIN-9, CIN-10, CIN-11, and CIN-12) shared similar views on how the situation worried them. CIN-1 shared her view about loss of confidence when she stated:

> *It is not easy at all. Sometimes we are psychologically down and lose confidence. That is why coming for review is sometimes difficult for us. We are only praying that, we will be able to overcome the situation.*

CIN-4 indicated a constant state of worrying as he added:

> *We get so worried sometimes. You see, you try to take your mind of it and concentrate on what you are doing. But as you try to focus, your attention is shifted to it from time to time.*

CIN-8 aired his views about being emotionally traumatized:

> *Emotionally we are down because what we are expecting, we are not getting. That put us off. We are worried as it seems things are delaying. This is not the first time we are seeking care but the situation is the same. Is like we should give up.*

The response from the interviewees show that they have diverse reasons for discontinuation of treatment at fertility clinics.

## Health practitioners' responses on why patients discontinue treatment

### Cost of treatment

One of the major reasons' why patients discontinued infertility treatment as indicated by the Key Informant Interviewees (health practitioners) is financial challenges. Almost all the interviewees mentioned financial difficulties as a common reason for discontinuation of infertility services. For example, HP-1 stated:

*As for the reasons, is mostly because of money issues. Lack of funds to pursue the investigations and treatment is one of the main reasons they discontinue treatment.*

Another account from HP-4 indicated:

*Most of them have financial challenges. The investigations and treatment are very expensive. This tend to put many of them off after they have started the whole process.*

Similarly, HP-5 also stated:

*They always complain of expensive investigations and cost of medications. That is why most of them discontinue with treatment*

HP-8 also reiterated his view when he said:

*Another reason is the cost of treatment. You know, national health insurance does not cover the services so the clients bear all the cost. Is something that can put them off.*

### Husbands' unsupportiveness

Another reason the health practitioners emphasized was lack of support from the male partners. Most of them (HP-1, HP-3, HP-4, HP-6) indicated that the male partners refuse to support the women in seeking care.

For instance, HP-1 said:

*Second one is unwillingness of the husbands to come and do investigation. Like you ask them to come for semen test and they don't show up. That is it.*

In a similar way, HP-6 also indicated:

*Some of the men don't really support the women. They refuse to come for the investigations. That is why most of them are not able to continue with treatment.*

HP-4 also recounted:

*Another reason is also lack of cooperation from male partners. They fail to come along with the women. The doctors are supposed to assess them together so if they don't show up, it demoralizes the women so they end up not coming again.*

### Unmet outcome

An important reason that caused clients to terminate care was a delayed outcome of treatment. Interviewees (HP-3, HP-6, HP-8) indicated that the delay in achieving expected outcome is a major factor that contributes to discontinuation of care.

For example, HP-3 stated:

*The main reason is because they are fed up. Since they don't see better result after coming for some time, they tend to give up. When this happens, they don't come again at all.*

HP-8 also shared his view when he indicated:

*What I can say is that most of them want instant results. But is a process so they easily give up. Most of them do not have the patience to wait for long. To some of them, the mere fact that they have come for treatment and within some few days they are not seeing results as expected, they tend to give up.*

In a similar way, HP-6 recounted:

*Some of them too, when they come and they don't achieve the results as expected, they give up. So not seeing desired results is an important factor that affects their compliance with health.*

### Long distance

Long distance also accounts for a common reason why patients discontinue treatment. Some of them have to travel from far places within the region, and sometimes beyond to seek care. Some of the respondents (HP-2, HP-7, HP-8) shared their views on how distance affects the way patients adhere to treatment.

For example, HP-2 stated:

*One of the main reasons I think clients don't come is because of distance. You know, some of them come all the way from Takoradi and other far places. That is one of the hinderances for them.*

HP-7 also said:

*I think distance might be a contributing factor. Some do come from far places. You know our clinics operates once in a week, so if by any reason, anything should come across them, they are not able to come as expected.*

In a similar way, HP-8 also shared his views when he indicated:

*The third reason is distance. No access to it where they live, especially those in rural areas. They have to travel all the way to the city to access these services since is not available over there.*

### Stigmatization

Stigmatization was one of the important reasons for discontinuation of infertility treatment, as shared by health professionals. Most of them do not feel comfortable when others get to know that they are seeking treatment for infertility. For instance,

HP-8 shared his view when he said:

*They feel stigmatized. They feel people will be asking them why they are always going to hospital, and what is wrong with them.*

Similarly, HP-5 also stated:

*What I can say is that, most of them don't always feel comfortable coming for treatment. They don't want some of the people close to them to know that, this is what they are suffering from. They think that others will be gossiping about them. This forces some of them to seek care at places far from where they live or work. As you might be aware, some of them already feel the pressure of social stigma.*

The above responses from health practitioners were shared as the common reason why infertile patients usually discontinue treatment.

## Discussion

The study was conducted to explore the views of patients and healthcare professionals on the reasons for discontinuation of fertility care by infertile patients prior to achieving conception. It analyses the data critically with reference to relevant literature in an attempt to explore deeper meanings of the responses, to unravel the issues and understand the phenomenon. Among the healthcare professionals interviewed were doctors and nurses from the four fertility centers who provided direct care to the patients. The respondents expressed diverse views on the factors that influenced discontinuation of infertility treatment in Ghana, which were not discovered in our previous study since it relied on secondary data [16]. It is evident from this qualitative study that patients who discontinued treatment cited various reasons to back their actions. Despite the major overlaps of responses, there were slight differences with regards to the reasons disclosed by the patients and the healthcare professionals. The findings revealed that, although various fertility clinics exist to ensure that the needs of infertile couples are met, there were inherent challenges that hindered patients from seeking treatment. Among the major challenges included the lack of cooperation from male partners, not achieving the expected outcome, adoption of alternative treatment, financial challenges, poor satisfaction with care, and relocation or lockdown as part of the COVID-19 restrictions. Given the psychological distress and stigma attached to infertility, and other negative socio-cultural consequences that infertile patients experience in Ghana [18], immediate intervention is required to rectify such challenges.

Several studies have attempted to understand why patients discontinue fertility treatment, with psychological burden cited as one of the most common reasons for discontinuation [2, 19]. However, in the current study, one of the major factors that influenced the patients to discontinue treatment, as recounted by patients, and healthcare professionals, was the non-cooperation of male partners. Such attitudes of the men create a problem during the assessment and evaluation of infertile couples, owing to the fact that men and women contribute approximately equally to the causes of infertility [20]. Despite these findings, in many parts of the world, both men and women are unlikely to attribute infertility to the male partner, a factor that is believed to contribute to the low patronage of men in seeking fertility treatment [21]. In sub-Saharan Africa, especially Ghana, there is a high social perception that women are responsible for infertility among couples [22]. This is true for most African countries, where there is a strong cultural perception and the belief that African men do not become infertile [23]. This perception makes most men feel relaxed about taking important steps towards seeking

treatment. It is important to know that failure of any of the partners to avail themselves for evaluation may hinder the entire treatment process and outcome. Given the fact that the diagnosis and treatment of infertility requires the cooperation of both partners, it remains imperative to put in appropriate measures that will address such challenges.

The uncertainty and perceived delay in treatment outcome also served as an important reason for discontinuation of fertility treatment. Expected treatment outcome (pregnancy and live birth), remains the ultimate desire of infertile patients undergoing pharmacotherapy for infertility. In Ghana for instance, new couples struggling with infertility are constantly pressured and stigmatized by their extended families and the community. This compels the majority of the couples to seek early treatment to avert the situation. However, the unmet expected outcomes for an ongoing treatment could easily result in apathy and discontinuation of treatment since they become psychologically distressed. This finding is in line with reports in existing literature, which indicate that the sense of futility associated with treatment [8, 24], and perceived poor prognosis and psychological dilemma [7], contribute to discontinuation of therapy. Although the evaluation and treatment of infertile patients may be cumbersome, and treatment outcome mostly depends on the cause of infertility, many patients do not see it that way. To most of them, a delayed outcome means a treatment failure, hence the need to seek alternative treatment. Usually, such couples begin to seek alternative treatments elsewhere, aimed at finding better and immediate solutions to their predicament. Extensive education prior to, and during the cause of treatment is very important to ensure that infertile patients comply with treatment.

It is therefore not surprising that most of the patients discontinued treatment with the intention of finding alternative solutions to their problem, as identified in the current study. Against the backdrop that infertility is a highly stigmatized and unacceptable condition in many communities in Ghana [22], seeking alternative treatment is a common practice among such patients. Numerous treatment options are being tried, including local concoctions, spiritual consultations, etc. It has become common practice to use local herbal medicines, as many native doctors claim to have an immediate cure for infertility. In line with this finding, widespread usage of local herbal medications among infertile couples has been reported in some parts of Africa [25, 26]. Similarly, it has been reported that some infertile women undergoing treatment abandon their treatment and resort to herbalists to drink some herbs in order to be cleansed from superstitious powers, which they perceive to be the cause of their fertility problems [27]. However, in most instances, seeking multiple treatment from different sources offer very little in terms of solving the problem. Therefore, extensive counseling of infertile couples, before and after commencing treatment is required, as most of them take such decisions out of desperation. Moreover, seeking fertility care in sub-Saharan Africa is very expensive due to the long-standing investigations and cost of medications. Hence, delayed, or unmet treatment outcome, may put further financial burden on such couples, which may subsequently cause them to discontinue treatment [28, 29].

Access to fertility clinics may also influence how infertile couples adhere to treatment [5]. In Ghana for instance, most of the fertility clinics are centralized in the major cities such as the regional capitals. Staying far from such cities, or relocation to places other than those major cities could make it difficult for them to access such services, as this may put further financial burden on the patients. Consequently, childless couples who are unable to travel to such areas to access fertility clinics usually resort to local herbal practitioners for treatment.

In view of the high number of infertility patients who have experienced failed treatments at fertility clinics, and the stress associated with seeking treatment, it has become imperative for fertility centers to meet the demands and primary needs of infertile couples. The current study revealed that some of the patients were not happy about the services they received at some

health facilities. This subsequently led to the discontinuation of treatment, and this is consistent with other findings about patients suffering from other conditions [30]. The perceived lack of experienced doctors was one of the major factors that affected their satisfaction with care, which was common in the state-owned facilities. Maintaining competent and experienced workers is necessary in meeting the needs of infertile couples who may already be traumatized emotionally.

Although stigmatization was mentioned by the health practitioners, the fertility patients scarcely voiced out anything regarding stigma as the cause of discontinuation of infertility treatment. It has been reported that, infertile women who involved themselves in seeking treatment experienced minimal levels of perceived stigma [31]. In the current study, all the respondents had availed themselves for treatment at some point in time, suggesting that such patients were less concerned about perceived stigma or infertility related stigma. Hence, not voicing it out as one of their major concerns. Another probable reason could be that some of the infertile patients, who might have discontinued their treatment due to perceived stigma, did not know how to voice it out, hence the complete omission of stigmatization from their responses.

Stigmatization, as recounted by the healthcare professionals, also contributes to discontinuation of infertility medical services. Children are highly valued in Ghana such that voluntary childlessness is not even welcomed. This compels many infertile couples to often seek early treatment before the general public become aware of their predicament. However, the use of some special places or days for fertility clinics make most infertile couples uncomfortable, as they may be easily tagged by some health professionals, as well as people within their community who may be aware of dates and times for such services [22]. More importantly, infertility in Ghana is usually managed by the obstetrician gynecologist at designated places within the hospital which are mostly overcrowded with a lot of female patients and nurses. Sometimes, in the midst of many women, one male patient is noticed sitting among them, so the reason for their presence at such places becomes so obvious. As such the environment may be intimidation and this may discourage most men from visiting fertility clinics, especially state-owned facilities. Adequate measures are therefore needed to provide standardized treatment procedures that will reduce exposure of clients to stigmatization.

## Strength and limitations of the study

Relatively few male partners agreed to participate in the study, even though their responses were much needed, given the fact that they were mostly uncooperative with treatment. However, the semi-structured interview guide enabled detailed information sharing which adequately brought out the key issues. Again, the health professionals who served as key informants provided useful information from another perspective that augmented the information provided by the patients.

## Conclusion

From the current study, patients and healthcare professionals shared both similar and diverse views on reasons for discontinuation of infertility treatment. Failure to address these factors could negatively impact treatment outcome, which may therefore worsen the psychological distress of infertile couples. Counselors of such couples are therefore required to engage both partners on the need for active participation in the treatment process in order to ensure positive outcome. Additionally, it will be prudent for fertility clinics to increase efforts at attracting and retaining highly qualified and experienced staff to enhance compliance with treatment.

## Supporting information

**S1 File.**
(DOCX)

## Acknowledgments

The authors wish to express their profound gratitude to all respondents who availed themselves for this study. We also wish to thank the management of the various fertility centers for making their facilities available for this research.

## Author Contributions

**Conceptualization:** Stephen Mensah Arhin, Kwesi Boadu Mensah, Charles Ansah.

**Data curation:** Evans Kofi Agbeno.

**Formal analysis:** Stephen Mensah Arhin, Isaac Tabiri Henneh, Eric Agyemang.

**Funding acquisition:** Kwesi Boadu Mensah, Evans Kofi Agbeno, Isaac Tabiri Henneh.

**Investigation:** Stephen Mensah Arhin, Eric Agyemang.

**Methodology:** Stephen Mensah Arhin, Kwesi Boadu Mensah, Evans Kofi Agbeno, Diallo Abdoul Azize, Isaac Tabiri Henneh, Eric Agyemang, Charles Ansah.

**Project administration:** Stephen Mensah Arhin, Kwesi Boadu Mensah.

**Resources:** Stephen Mensah Arhin, Kwesi Boadu Mensah, Isaac Tabiri Henneh.

**Software:** Isaac Tabiri Henneh.

**Supervision:** Kwesi Boadu Mensah, Charles Ansah.

**Validation:** Evans Kofi Agbeno, Eric Agyemang.

**Visualization:** Stephen Mensah Arhin, Kwesi Boadu Mensah, Evans Kofi Agbeno, Diallo Abdoul Azize, Isaac Tabiri Henneh, Charles Ansah.

**Writing – original draft:** Stephen Mensah Arhin, Kwesi Boadu Mensah, Eric Agyemang, Charles Ansah.

**Writing – review & editing:** Stephen Mensah Arhin, Kwesi Boadu Mensah, Evans Kofi Agbeno, Diallo Abdoul Azize, Isaac Tabiri Henneh, Eric Agyemang, Charles Ansah.

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
