## [Decision Letter · Decision Letter 0]

7 Jun 2022

PONE-D-22-12329Pharmacotherapy of infertility in Ghana: Why do subfertile couples discontinue their fertility treatment?PLOS ONE

Dear Dr. Arhin,

Thank you for submitting your manuscript to PLOS ONE. After careful consideration, we feel that it has merit but does not fully meet PLOS ONE’s publication criteria as it currently stands. Therefore, we invite you to submit a revised version of the manuscript that addresses the points raised during the review process. In addition to comments provided by the two reviewers, the please consider the following:

Line 26: Consider revision to "The study employed an exploratory qualitative design"

Line 26-26: It is enough to say "purposive sampling technique"

Line 28: Write ‘8’ in word

Line 29: Consider changing "were sampled and interviewed" to just "were interviewed"

In the abstract, state the data analysis approach in a short sentence.

Line 32: Consider revising to: “The major reasons for discontinuation of infertility medical services included….”

Line 37-39: It might be better to say the insight provided rather that say it will provide insight. The current statement sounds like a justification for the study, not a recommendation from the findings, as might be the intention of the authors.

Line 94: This initial statement does not make your study population clear. Twenty couples ordinarily means 20 pairs of persons, making 40 persons. Also, please indicate that you did not interview both members of each couple pair, and why not.

Please ensure to remove any personal identifying information from your manuscript, including names of organizations where identifying them is not essential to the information being provided. You must justify the inclusion of names that identify any entity.

Concerning your suggestions on the approach to data collection, infertility is a sensitive issue and FGD may still not be the most appropriate fora for data collection, particularly when the proposed participants are currently experiencing infertility.

In line with the comments of one of the reviewers, consider removing the word "subfertile" from your title.

Do pay attention to typographical and grammatical errors.

We look forward to receiving your revised manuscript.

Kind regards,

Olujide Arije

Academic Editor

PLOS ONE

Journal Requirements:

Reviewers' comments:

Reviewer's Responses to Questions

**Comments to the Author**

1. Is the manuscript technically sound, and do the data support the conclusions?

Reviewer #1: Yes

Reviewer #2: Yes

2. Has the statistical analysis been performed appropriately and rigorously? 

Reviewer #1: N/A

Reviewer #2: N/A

3. Have the authors made all data underlying the findings in their manuscript fully available?

Reviewer #1: Yes

Reviewer #2: No

4. Is the manuscript presented in an intelligible fashion and written in standard English?

Reviewer #1: No

Reviewer #2: Yes

5. Review Comments to the Author

Reviewer #1: This is an interesting paper which focuses specifically on pharmacotherapy for sub-fertile couples in Ghana and why these couples may discontinue treatment before achieving pregnancy.

In the title you talk about choices that subfertile couples make, but I am not sure how representative the term couples is of the study population?

Some small changes which would make some things clearer for me:

line 72, pg 4 - change to 'for infertility' instead of 'of infertility'

line 73, pg 4 - add Metropolis to Cape Coast, as you later refer to just 'the metropolis' this confused me, and then you later call it Cape Coast Metropolis

line 76, pg 4 - to enable stakeholders 'to' develop (word missing)

lines 94-96, pg 5 - I would like to see this more clearly, on first reading it appears you interviewed 20 couples, but you then say you interviewed 14 women and 6 men. Maybe be specific that you interviewed 20 clients, which were a combination of 14 women and 6 men - but were any of these couples?

line 130 - pg 7 - who did the transcription? and also did you use a specific type of thematic analysis?

line 132, pg 7 - reword 'two of the authors coded few' this reads as though they coded a minimal amount - might be useful to say how many and which (HCP or clients)

line 135, pg 7 - It feels more accurate to say that themes were identified than that they emerged.

line 138, pg 8 - I think you mean that you were maintaining confidentiality, the opposite to disclosing the identity of the participants?

line 190, pg 10 - 'live' birth (also line 430)

line 433, pg 21 - you talk about internal and external pressures - which do you mean specifically?

line 485, pg 24 - midst, not mist

I really enjoyed reading this, thank you. And I think you have made some really important points, particularly about the abdication of responsibility of the male partners.

There has been much made of the 'delay' to treatment outcomes - do you have a sense, or data to indicate what sort of time frames that are contributing to the drop out of treatment?

I think that this article may need some editing for language and clarity, but the data support the conclusions and suggestions you have made for how services could be improved.

Reviewer #2: Review PONE-D-22-12329

This a very important qualitative interview study on why couples in Ghana discontinue fertility treatment (pharmacotherapy). A topic where we have very limited knowledge from low/middle income countries. Globally, infertility is one of the most frequent diseases among reproductive-aged women and men. In middle and low-income countries the access to fertility treatment is limited and is not affordable for many couples with infertility. The study is carefully conducted, data clearly presented and the manuscript is easy to follow.

I have a number of minor suggestions –

Terminology

I suggest to follow the terminology in the International Glossary on Infertility and Fertility Care, 2017 (Zegers-Hochschild et al., Human Reproduction, 2017). My suggestion is to use “infertility” as this is clearly defined (and not “subfertility” that is usually not well-defined); fertility care, fertility clinics, fertility treatment and in line with this fertility patients.

Throughout the manuscript is used both “clients” and “fertility patients”. I suggest to use only one term (fertility patients). Similarly, is used both “infertility” and “subfertility”. I suggest infertility as stated above.

Introduction

is well-written and provides a good overview. I suggest to include a few more papers on infertility in Sub-Saharan countries. Search e.g., for papers in Pubmed published by Silke Dyer, Johanne Sundby, Florence Naab and others and you will find papers e.g., on access to fertility care, the psychosocial impact of infertility, studies on the impact of male infertility, how fertility patients assess treatment. Some could probably also be included in the Discussion section.

Materials and methods

I suggest to add either a table or a short overall description of the study participants (besides their gender). For the fertility patients – e.g., age range, previous reproductive history (have child/children or not), previous treatment. For the health care professionals – e.g, number of years in fertility care. This info should be provided on a group level, so the study participants could not be identified.

I suggest to put the paragraph on Ethical approval as the last section before Results.

Please, provide a reference to the thematic analysis used.

The description of how you assigned codes to the interview transcripts is not a Result and should be placed in the M&M section.

Results

I suggest you provide a table showing themes from interviews with fertility patients and with health care professionals, respectively. To help the reader to have an overview.

Otherwise the result section is easy to follow and quotes selected are illustrative for the theme.

In quotes, I would recommend not to mention e.g., name of fertility clinics in order not to reveal study participant identity.

I would prefer Psychological distress (p. 15 , line 299) instead of disturbances (which sounds closer to psychiatric disorders).

Discussion is relevant. I found it interesting that the fertility patients’ do not mentioning stigma as a reason for discontinuation, but it was mentioned by the health care professionals. Could this difference be discussed in short?

6. PLOS authors have the option to publish the peer review history of their article (what does this mean?). If published, this will include your full peer review and any attached files.

Reviewer #1: **Yes: **Eleanor Molloy

Reviewer #2: No

---

## [Author Response · Author response to Decision Letter 0]

19 Jul 2022

RESPONSE TO EDITOR'S AND REVIEWERS' COMMENTS

Editor’s comments

Thank you for submitting your manuscript to PLOS ONE. After careful consideration, we feel that it has merit but does not fully meet PLOS ONE’s publication criteria as it currently stands. Therefore, we invite you to submit a revised version of the manuscript that addresses the points raised during the review process.

In addition to comments provided by the two reviewers, the please consider the following:

Comment

Line 26: Consider revision to "The study employed an exploratory qualitative design"

Response

The change has been effected in the revised manuscript. (line 26, pg 2).

Comment:

Line 26-27: It is enough to say "purposive sampling technique" 

Response

The change has been effected in the new manuscript (line 26, pg 2).

Comment

Line 28: Write ‘8’ in word

Response:

The number ‘8’ has been written in full in the new manuscript (line 28, pg 2)

Comment:

Line 29: Consider changing "were sampled and interviewed" to just "were interviewed"

Response:

The change has been effected in the revised manuscript (line 29, pg 2).

Comment:

In the abstract, state the data analysis approach in a short sentence.

Response: 

The data analysis approach has been mentioned in the abstract in the revised manuscript (line 31).

Comment 

Line 32: Consider revising to: “The major reasons for discontinuation of infertility medical services included….”

Response

The change has been effected in the revised manuscript (line 32).

Comment 

Line 37-39: It might be better to say the insight provided rather that say it will provide insight. The current statement sounds like a justification for the study, not a recommendation from the findings, as might be the intention of the authors.

Response

The change has been effected in the revised manuscript (line 37-38).

Comment

Line 94: This initial statement does not make your study population clear. Twenty couples ordinarily means 20 pairs of persons, making 40 persons. 

Response

The term ‘couples’ has been changed to ‘patients’ in the revised manuscript.

Comment:

Also, please indicate that you did not interview both members of each couple pair, and why not.

Response:

None of the 20 respondents were from the same couple pair. The telephone calls were made to the number(s) provided by the couple-pair during their first visit. However, any of the partners who was available and willing to participate in the study was recruited. 

Comment 

Please ensure to remove any personal identifying information from your manuscript, including names of organizations where identifying them is not essential to the information being provided. You must justify the inclusion of names that identify any entity.

Response 

Names of organizations that are not essential to the information being provided have been removed from the revised manuscript. 

Comment 

Concerning your suggestions on the approach to data collection, infertility is a sensitive issue and FGD may still not be the most appropriate fora for data collection, particularly when the proposed participants are currently experiencing infertility.

Response

Thank you for your comment regarding the approach to data collection we suggested as recommendation for further research. The suggestion of FGD has been removed from the revised manuscript.

Comment

In line with the comments of one of the reviewers, consider removing the word "subfertile" from your title.

Response

The term “subfertile” has been removed from the title of the revised manuscript.

Comment

Do pay attention to typographical and grammatical errors.

Response

The authors are grateful to the Editor for his constructive comments and suggestions in reshaping out manuscript.

The revised manuscript has been proofread by people and typographical and grammatical errors.

RESPONSE TO REVIEWER 1

Comment:

In the title you talk about choices that subfertile couples make, but I am not sure how representative the term couples is of the study population?

Response:

It is true that the female respondents were more than the male respondents. Throughout the study, it was observed that the males were less forthcoming when it comes to cooperating with treatment and willingness to participate in the study. This has largely been attributed to the cultural setting in African, which places the responsibility of childbirth on the doorsteps of women. This has been well-elaborated in the manuscript. As such, we have revised the term “couples” to “patients” in the title.

Comment:

Some small changes which would make some things clearer for me:

line 72, pg 4 - change to 'for infertility' instead of 'of infertility'

Response:

The change has been effected (line 74, pg 4).

Comment:

line 73, pg 4 - add Metropolis to Cape Coast, as you later refer to just 'the metropolis' this confused me, and then you later call it Cape Coast Metropolis

Response:

The change has been effected (line 82, page 4).

Comment:

line 76, pg 4 - to enable stakeholders 'to' develop (word missing)

Response:

The change has been effected (line 78, page 4).

Comment:

lines 94-96, pg 5 - I would like to see this more clearly, on first reading it appears you interviewed 20 couples, but you then say you interviewed 14 women and 6 men. Maybe be specific that you interviewed 20 clients, which were a combination of 14 women and 6 men - but were any of these couples?

Response:

The suggestion has been well noted and correction effected in the revised manuscript. Please refer to lines 96-97 on page 5. 

On the question regarding whether any of the respondents were couples, the telephone calls were made to the number(s) provided by the couples during their first visit. However, any of the partners who was willing to participate in the study was recruited. As such, none of the 20 respondents were from the same couple pair. 

Comment:

line 130 - pg 7 - who did the transcription? and also did you use a specific type of thematic analysis?

Response

Two of the authors inductively coded four of the transcripts (two from CIN and two from HP) independently. The initial codes were review6d by three of the authors for consistency before a consensus was reached on the codes and the coding system. The first author then coded all the transcripts using the coding system agreed upon (line 145-149, pg 8).

On the question of a specific type of thematic analysis, a thematic content analysis was used to analyze the data (line 143-144, pg 8).

Comment:

line 132, pg 7 - reword 'two of the authors coded few' this reads as though they coded a minimal amount - might be useful to say how many and which (HCP or clients).

Response:

Two of the authors inductively coded four of the transcripts (two from patients and two from the healthcare personnel). As such the changes have been effected in the revised manuscript. (lines 146-147).

Comment:

line 135, pg 7 - It feels more accurate to say that themes were identified than that they emerged.

Response:

We have changed the term “emerged” to “identified” in the revised manuscript (line 151, pg 8).

Comment:

line 138, pg 8 - I think you mean that you were maintaining confidentiality, the opposite to disclosing the identity of the participants?

Response:

The correction has been effected in the revised manuscript. (Please refer to page 8, line 134).

Comment:

line 190, pg 10 - 'live' birth (also line 430)

Response:

The correction has been effected. (line 212, page 12).

Comment:

line 433, pg 21 - you talk about internal and external pressures - which do you mean specifically?

Response:

The sentence has been clarified in the revised manuscript (line 457-458, pg 23).

Comment:

line 485, pg 24 - midst, not mist

Response:

The word has been spelt correctly in the revised manuscript. Line 518, page 27.

Comment

I really enjoyed reading this, thank you. And I think you have made some really important points, particularly about the abdication of responsibility of the male partners.

There has been much made of the 'delay' to treatment outcomes - do you have a sense, or data to indicate what sort of time frames that are contributing to the drop out of treatment?

I think that this article may need some editing for language and clarity, but the data support the conclusions and suggestions you have made for how services could be improved.

Response

Authors are grateful to the reviewer for her kind comments about our manuscript. We also thank her for the constructive comments which have made the revised manuscript better.

From the observations made in the study as well as responses from health professionals, a delay in expected outcome for about 1–3 months for males and 2-4 months for the females, usually caused them to abdicate treatment. We have included the time frames in the revised manuscript. 

The manuscript has been proofread by people with English as their first language.

Reviewer #2: Review PONE-D-22-12329

This a very important qualitative interview study on why couples in Ghana discontinue fertility treatment (pharmacotherapy). A topic where we have very limited knowledge from low/middle income countries. Globally, infertility is one of the most frequent diseases among reproductive-aged women and men. In middle and low-income countries the access to fertility treatment is limited and is not affordable for many couples with infertility. The study is carefully conducted, data clearly presented and the manuscript is easy to follow.

I have a number of minor suggestions.

Terminology

I suggest to follow the terminology in the International Glossary on Infertility and Fertility Care, 2017 (Zegers-Hochschild et al., Human Reproduction, 2017). My suggestion is to use “infertility” as this is clearly defined (and not “subfertility” that is usually not well-defined); fertility care, fertility clinics, fertility treatment and in line with this fertility patients.

Throughout the manuscript is used both “clients” and “fertility patients”. I suggest to use only one term (fertility patients). Similarly, is used both “infertility” and “subfertility”. I suggest infertility as stated above.

Response:

We thank the reviewer for drawing our attention to this. We have replaced “subfertility” with “infertility” in the revised patients.

INTRODUCTION

Comment

Is well-written and provides a good overview. I suggest to include a few more papers on infertility in Sub-Saharan countries. Search e.g., for papers in Pubmed published by Silke Dyer, Johanne Sundby, Florence Naab and others and you will find papers e.g., on access to fertility care, the psychosocial impact of infertility, studies on the impact of male infertility, how fertility patients assess treatment. Some could probably also be included in the Discussion section. 

Response 

The recommended papers have been read as suggested. The introduction and the discussion sections have been updated accordingly.

MATERIALS AND METHODS

Comment 

I suggest to add either a table or a short overall description of the study participants (besides their gender). For the fertility patients – e.g., age range, previous reproductive history (have child/children or not), previous treatment. For the health care professionals – e.g, number of years in fertility care. This info should be provided on a group level, so the study participants could not be identified.

Response 

Tables indicating the overall description of study participants have been provided in the revised manuscript.

Comment 

I suggest to put the paragraph on Ethical approval as the last section before Results. 

Response 

The changes have been effected in the revised manuscript

Comment

Please, provide a reference to the thematic analysis used. 

Response 

Reference has been provided in the revised manuscript.

Comment 

The description of how you assigned codes to the interview transcripts is not a Result and should be placed in the M&M section.

Response

The changes have been effected as suggested. 

RESULTS

I suggest you provide a table showing themes from interviews with fertility patients and with health care professionals, respectively. To help the reader to have an overview.

Otherwise the result section is easy to follow and quotes selected are illustrative for the theme.

Comment

In quotes, I would recommend not to mention e.g., name of fertility clinics in order not to reveal study participant identity.

Response 

Names of fertility clinics have been removed from the revised manuscript.

Comment

I would prefer Psychological distress (p. 15 , line 299) instead of disturbances (which sounds closer to psychiatric disorders).

Response 

The correction has been effected in the revised manuscript (line 323, page 18).

Comment 

Discussion is relevant. I found it interesting that the fertility patients’ do not mentioning stigma as a reason for discontinuation, but it was mentioned by the health care professionals. Could this difference be discussed in short?

Response

The discussion on sigma has been updated in the revised manuscript (line 501-509, page 26).

---

## [Decision Letter · Decision Letter 1]

11 Aug 2022

PONE-D-22-12329R1Pharmacotherapy of infertility in Ghana: Why do infertile patients discontinue their fertility treatment?PLOS ONE

Dear Dr. Arhin,

Thank you for submitting your manuscript to PLOS ONE. After careful consideration, we feel that it has merit but does not fully meet PLOS ONE’s publication criteria as it currently stands. Therefore, we invite you to submit a revised version of the manuscript that addresses the points raised during the review process.

Please see "additional Editor comments" section below

We look forward to receiving your revised manuscript.

Kind regards,

Thomas Tischer

Staff Editor

PLOS ONE

Journal Requirements:

Additional Editor Comments (if provided):We noticed that your study included 20 participants, however the "occupation" row in Table 1 adds up to 21 individuals. If it was possible to provide multiple answers to this questions, please indicate this. Otherwise, please ensure to correct the "occupation" row with the correct valueWe noticed that in Table 1 there is a difference in the capitalization of "Male" vs "female"Please check your other tables for similar inconsistencies (numbers and grammar)

Reviewers' comments:

Reviewer's Responses to Questions

**Comments to the Author**

1. If the authors have adequately addressed your comments raised in a previous round of review and you feel that this manuscript is now acceptable for publication, you may indicate that here to bypass the “Comments to the Author” section, enter your conflict of interest statement in the “Confidential to Editor” section, and submit your "Accept" recommendation.

Reviewer #1: All comments have been addressed

Reviewer #2: All comments have been addressed

2. Is the manuscript technically sound, and do the data support the conclusions?

Reviewer #1: Yes

Reviewer #2: Yes

3. Has the statistical analysis been performed appropriately and rigorously? 

Reviewer #1: N/A

Reviewer #2: N/A

4. Have the authors made all data underlying the findings in their manuscript fully available?

Reviewer #1: Yes

Reviewer #2: No

5. Is the manuscript presented in an intelligible fashion and written in standard English?

Reviewer #1: Yes

Reviewer #2: Yes

6. Review Comments to the Author

Reviewer #1: Thank you for the work you have done on this manuscript, it is an important field of study and crucial that we are exploring experiences of accessing healthcare across all populations and countries.

Reviewer #2: The authors have carefully revised the manuscript in accordance with comments from reviewers and AE. I have two additional minor comments:

1. Most places the author states "infertile patients" and fewer places "fertility patients" (e.g., Table 3, and in sub-heading line 205). I suggest to be consistent throughout the manuscript and use one term only. As previously mentioned, I prefer "fertility patients"; however, this is not a requirement.

2. I would avoid the term "drop-out" of fertility treatment, as "drop-out" could indicate something that just happened passively. Studies - including your own - clearly indicates that fertility patients who discontinue treatment has been actively through a decision-making process.

On p. 6, Sampling, I suggest e.g. to revise to "20 fertility patients who had discontinued treatment " and in Table 1 to change "drop-out time" to Treatment discontinuation. I am not sure whether it is discontinuation after 1-3 months of being in fertility treatment?

7. PLOS authors have the option to publish the peer review history of their article (what does this mean?). If published, this will include your full peer review and any attached files.

Reviewer #1: **Yes: **Eleanor Molloy

Reviewer #2: No

---

## [Author Response · Author response to Decision Letter 1]

22 Aug 2022

Response to editor’s and reviewers’ comments

We are very much grateful to the editor and the reviewers for their meaning contributions towards reshaping of our manuscript. 

Response to Editor’s comment

Comment: We noticed that your study included 20 participants, however the "occupation" row in Table 1 adds up to 21 individuals. If it was possible to provide multiple answers to this question, please indicate this. Otherwise, please ensure to correct the "occupation" row with the correct value.

Response: This was an error, and correction has been effected in the revised manuscript. 

Comment: We noticed that in Table 1 there is a difference in the capitalization of "Male" vs "female"

Response: Appropriate changes have been made to ensure consistency in the revised manuscript. 

Comment: Please check your other tables for similar inconsistencies (numbers and grammar).

Response: Other tables have been checked and necessary changes effected in the revised manuscript.

Response to reviewer 2 comments

Comment: Most places the author states "infertile patients" and fewer places "fertility patients" (e.g., Table 3, and in sub-heading line 205). I suggest to be consistent throughout the manuscript and use one term only. As previously mentioned, I prefer "fertility patients"; however, this is not a requirement.

Response: We are grateful to you for drawing our attention to this error. The necessary corrections have been effected to ensure uniformity and consistency. However, the term ‘infertile patients’ has been maintained in the revised manuscript. 

Comment: I would avoid the term "drop-out" of fertility treatment, as "drop-out" could indicate something that just happened passively. Studies - including your own - clearly indicates that fertility patients who discontinue treatment has been actively through a decision-making process.

On p. 6, Sampling, I suggest e.g. to revise to "20 fertility patients who had discontinued treatment " and in Table 1 to change "drop-out time" to Treatment discontinuation. I am not sure whether it is discontinuation after 1-3 months of being in fertility treatment?

Response: The suggested corrections have been effected accordingly in the revised manuscript.

On the issue of discontinuation, the 1-3 months means the average duration after which the patients discontinued their treatment.

---

## [Editor Report · Decision Letter 2]

1 Sep 2022

Pharmacotherapy of infertility in Ghana: Why do infertile patients discontinue their fertility treatment?

PONE-D-22-12329R2

Dear Dr. Arhin,

We’re pleased to inform you that your manuscript has been judged scientifically suitable for publication and will be formally accepted for publication once it meets all outstanding technical requirements.

Kind regards,

George Vousden

Staff Editor

PLOS ONE
---

## [Editor Report · Acceptance letter]

7 Oct 2022

PONE-D-22-12329R2 

Pharmacotherapy of infertility in Ghana: Why do infertile patients discontinue their fertility treatment? 

Dear Dr. Arhin:

I'm pleased to inform you that your manuscript has been deemed suitable for publication in PLOS ONE. Congratulations! Your manuscript is now with our production department. 

Kind regards, 

on behalf of

Dr. George Vousden 

Staff Editor

PLOS ONE